# The SP19 chronology for the South Pole Ice Core - Part 2: gas chronology, Δage, and smoothing of atmospheric records

Jenna A. Epifanio[1], Edward J. Brook[1], Christo Buizert[1], Jon S. Edwards[1], Todd A. Sowers[2], Emma C. Kahle[3], Jeffrey P. Severinghaus[4], Eric J. Steig[3], Dominic A. Winski[5,6], Erich C. Osterberg[7], Tyler J. Fudge[3], Murat Aydin[8], Ekaterina Hood[9], Michael Kalk[1], Karl J. Kreutz[5,6], David G. Ferris[7], and Joshua A. Kennedy[10]

[1]College of Earth, Ocean, and Atmospheric Sciences, Oregon State University, Corvallis OR, USA
[2]The Earth and Environmental Systems Institute, Pennsylvania State University, University Park, PA, USA
[3]Department of Earth and Space Science, University of Washington, Seattle WA, USA
[4]Scripps Institution of Oceanography, University of California San Diego, La Jolla CA, USA
[5]School of Earth and Climate Sciences, University of Maine, Orono, Maine, USA
[6]Climate Change Institute, University of Maine, Orono, Maine, USA
[7]Department of Earth Sciences, Dartmouth College, Hanover, NH, USA
[8]Department of Earth System Sciences, University of California, Irvine, CA, USA
[9]Department of Geology and Environmental Earth Sciences, University of Miami, FL, USA
[10]Department of Chemistry and Biochemistry, South Dakota State University, Brookings, South Dakota, USA

*Correspondence to*: Jenna A. Epifanio (epifanij@oregonstate.edu)

**Abstract.** A new ice core drilled at the South Pole provides a 54,000-year paleoenvironmental record including the composition of the past atmosphere. This paper describes the SP19 chronology for the South Pole atmospheric gas record and complements a previous paper (Winski et al., 2019) describing the SP19 ice chronology. The gas chronology is based on a discrete methane ($CH_4$) record with 20- to 190-year resolution. To construct the gas time scale abrupt changes in atmospheric $CH_4$ during the glacial period and centennial $CH_4$ variability during the Holocene were used to synchronize the South Pole gas record with analogous data from the West Antarctic Ice Sheet Divide ice core. Stratigraphic matching based on visual optimization was verified using an automated matching algorithm. The South Pole ice core recovers all expected changes in $CH_4$ based on previous records. Gas transport in the firn results in smoothing of the atmospheric gas record with a smoothing function spectral width that ranges from 30 to 78 years, equal to 3% of the gas age-ice age difference, or Δage. The new gas chronology, in combination with the existing ice age scale from Winski et al. (2019), allows a model-independent reconstruction of the gas age-ice age difference through the whole record, which will be useful for testing firn densification models.

## 1 Introduction

Ice core records provide detailed reconstructions of past climate in the polar regions and unique global records of the past atmosphere. These records are of high resolution and are well-dated, allowing comparisons to events in other ice cores and paleoarchives (Buizert et al., 2015; Elderfield et al., 2012; Hodell et al., 2017; Marcott et al., 2013). The recently collected South Pole ice core (SPC14) expands a spatial array of ice cores drilled across Antarctica that extend into the last glacial period.

SPC14 is an intermediate depth (1751 m) ice core that was drilled as a part of the South Pole ice core (SPICEcore) project and was collected in the 2014/15 and 2015/16 field seasons (Souney et. al., 2020). The core provides ice and gas data through part of last glacial period, to 54,302 years before present (BP, with 0 BP = 1950 CE; Winski et al., 2019). Because drilling stopped almost 1000 m above bedrock, folding and mixing of layers at the bottom of the core is not a concern, resulting in a stratigraphically

continuous record for the entire length of the core. The core location is 89.99˚ S, 98.16˚ W, at surface elevation of 2835 m on the polar plateau of the East Antarctic ice sheet. The current annual accumulation rate is 8 cm/a water equivalent, (Lilien et al., 2018; Mosley-Thompson et al., 1999) with an annual-mean temperature of -51˚ C as measured in the firn (Severinghaus et al., 2001). Due to its geographic location, ice accumulating at the site has low levels of trace impurities such as black carbon (BC), major ions, dust, and trace elements (Casey et al., 2017). These characteristics are an advantage for the measurements of ultra-trace gases

such as ethane, methyl chloride, and methyl bromide (Aydin et al., 2004; Lee et al., 2020b; Nicewonger et al., 2018; Saltzman et al., 2004), one of the primary goals of the SPICEcore project.

Air permeates through the porous firn layer and is trapped at depth between 50-120 m, depending on local climate. As a result, air is younger than ice at the same depth (Schwander and Stauffer, 1984). This gas age-ice age difference (∆age) can range from tens to thousands of years, depending on the site ice accumulation rate and temperature. Gas chronologies for previously

collected ice cores have been created either through calculating gas ages using an existing ice chronology coupled with models of ∆age, or by stratigraphically matching features in gas records with previously dated records in other cores (Blunier et al., 2007; Buizert et al., 2014; J. Schwander and B. Stauffer, 1984; Petit et al., 1999; Schwander et al., 1997; Sowers et al., 1992). Later Antarctic chronologies introduced constraints on gas age scales using $\delta^{15}N$, $\delta^{18}O_{atm}$ and firn models (Bazin et al., 2012; Veres et al., 2013). In cold, relatively low, accumulation rate sites similar to the South Pole, ∆age model uncertainty can be a major

contributor to the overall uncertainty in the gas age time scale. At the South Pole site today, ∆age at the bubble close-off depth is about 1000 years, large enough that the classical approach to calculating ∆age using a firn-densification model, typically having an uncertainty of about 20%, is insufficient for dating the gas at the precision needed to compare leads and lags between abrupt climate signals recorded in the ice core.

This paper focuses on the creation of a $CH_4$-based gas chronology for SPC14 using a stratigraphic matching approach and

is a companion to a paper describing the first ice chronology for the core (Winski et al., 2019). The gas and ice chronologies are collectively referred to as the SP19 chronology. $CH_4$ is a well-mixed atmospheric trace gas exhibiting globally synchronous abrupt variations on decadal to millennial scales (Blunier and Brook, 2001; Brook et al., 1996; Lee et al., 2020a; Rhodes et al., 2015b), making it an ideal choice for stratigraphic matching to existing ice core records. The chronology presented here relies on correlating $CH_4$ variations between the SPC14 and the West Antarctic Ice Sheet Divide (WD) ice cores, using millennial-scale abrupt

variations during the last ice age and glacial-interglacial transition, and centennial-scale variations during the Holocene. The WD2014 ice core chronology was created in two parts From 0-31.2 ka BP it is based on annual-layer counting; from 31.2-67.8 ka BP it is based on stratigraphic matching of WD $CH_4$ to NGRIP (North Greenland Ice core Project) $\delta^{18}O$, using a linear correction of the NGRIP age scale to improve the fit to the Hulu speleothem record (Buizert et al., 2015; Sigl et al., 2016). For both segments the WD2014 ∆age estimate is based on an $\delta^{15}N$-constrained firn densification model simulation.


We first describe relevant attributes of the SPC14 core and acquisition of the SPC14 CH$_4$ record, and then we discuss synchronization and optimization of the gas chronology. We also discuss key observations from our results, including implications for the Δage, smoothing of atmospheric gas records by gas transport in the South Pole firn, and short-term variability in atmospheric CH$_4$.

## 2 Methods

### 2.1 CH$_4$ measurements

High-resolution CH$_4$ concentration measurements were made along the entire length of SPC14, jointly at Oregon State University (OSU) and Pennsylvania State University (PSU). Samples from the 139 m - 1077 m interval were measured at PSU and samples in the 1078 - 1751 m interval were measured at OSU. Both labs measured samples in the 330 – 840 m and 1130 - 1150 m intervals for intercalibration. A correction of 6 parts per billion (ppb) were added to PSU CH$_4$ concentrations to correct for an offset that was revealed by the intercalibration measurements. A total of 2318 measurements (733 at PSU and 1598 at OSU) were

made on samples at 1067 individual depths resulting in 1 to 2-meter depth resolution throughout the entire core. Samples were measured in duplicate (832 depths), triplicate (46 depths), or in quadruplicate for the purpose of laboratory intercalibration (109 depths). 80 sample depths were measured without replication due to limited sample size or samples that had been broken during shipment.

       CH$_4$ concentrations measured at OSU were made using a wet-extraction technique as described in Grachev et al. (2009),

with updates by Mitchell et al. (2011) and Lee et al. (2018). Briefly, subsamples of the main core measuring 10 cm x 6 cm x 2.5 cm (with the 10 cm dimension oriented parallel to the vertical axis of the core) were split into replicate samples by cutting along the vertical axis. Each individual sample was then placed in a glass vacuum flask and attached to an automated analytical setup. The samples were kept frozen by immersing the flasks in an ethanol bath at -68 ˚C. After evacuating atmospheric air from the flasks with a vacuum pump, the flasks were submerged in a warm water bath for 30 minutes, melting the ice samples and releasing

the trapped air. The water was then refrozen, equilibrating to the temperature of the ethanol bath, over a period of 1 hour. Once the temperature of the flasks stabilized to -68 ˚C, air in the head space of each flask was expanded four times into a gas chromatograph (GC) for CH$_4$ analysis. Concentrations were quantified by comparison to a calibrated air standard at the beginning and end of each day (500.22 ppb for samples measured in 2016 and 481.25 ppb for any samples measured after 2016) on the NOAA04 CH$_4$ concentration scale (Dlugokencky et al., 2005).

Several corrections were made to the raw CH$_4$ concentration value measured at OSU including adjustments for a small quantity of CH$_4$ that remains dissolved in the melt water (Mitchell et al., 2013; Lee et al., 2019). Because gases do not reach complete solubility equilibrium during the melt-refreeze process, an empirical solubility correction is employed for this purpose. Mitchell et al., (2013) describes the experimental derivation of the correction. The derivation was repeated for SPC14 samples resulting in a correction factor of 1.7% (all sample concentrations measured at OSU were corrected for solubility by increasing the

measured value by 1.7%).

A small amount of additional $CH_4$ can also be present in measured samples due to the influence of air leaks or other contaminations. To quantify this blank correction at OSU, air-free ice (AFI) was routinely measured in conjunction with samples. Production of AFI is described by Mitchell et. al (2013). AFI was processed for analysis and measured in the same way as sample ice; however, an amount of standard air with a known mole fraction of $CH_4$ was added to the flask with AFI prior to the melt-refreeze step. Blank corrections derived from these measurements, also corrected for solubility effects, were subtracted from the measured concentration of each sample. Because samples were measured at different times, a blank correction was applied to each group of samples depending on when they were measured. Average blank corrections ranged from 6.6 ppb to 9.8 ppb for all samples measured at OSU. Data and information about corrections are provided in the supplementary material.

PSU $CH_4$ measurements (depths 139 m – 1077 m) were also made using an automated melt-refreeze method similar to the OSU system. However, the PSU system uses stainless steel flasks, which introduces an additional blank correction associated with $CH_4$ outgassing. The blank was estimated by analysing ice of a known $CH_4$ concentration through multiple melt-refreeze cycles. A regression between the excess $CH_4$ and the number of melt-refreeze cycles was completed to arrive at an estimate of 35 ±19 ppb blank correction. This correction was applied to all PSU samples. Further description of the PSU method can be found in WAIS Divide Project Members (2013).

All $CH_4$ concentrations are slightly affected by fractionation in the firn column due to gravity (Craig et al., 1988; Mitchell et al., 2011; Schwander et al., 1997; Sowers et al., 1992). The amount of gravitational fractionation is controlled by the thickness of the diffusive zone in the firn column, and can be estimated using the $\delta^{15}N$ of $N_2$. We corrected all measured $CH_4$ concentrations for gravitational fractionation following Mitchell et al. (2011) by interpolating the $\delta^{15}N$ of $N_2$ reported in Winski et al. (2019) to the depths of the $CH_4$ samples, and then using the relationship:

$$CH_{4corr} = CH_{4\,meas} \times \left(1 + \Delta M \frac{\partial^{15}N}{1000}\right) \qquad (1)$$

where $\Delta M$ is 12.92 g/mol, the difference in molecular weight between air (M = 28.96 g/mol) and the mass of the $CH_4$ (M = 16.04 g/mol). $\delta^{15}N$ varies from 0.63‰ to 0.46‰, with a mean value of 0.54‰. The correction ranges from 2.7 to 5.7 ppb (1 $\sigma$ = 0.83 ppb).

The SPC14 discrete $CH_4$ record, measured jointly at OSU and PSU, spans the period 130 years to 52,482 years BP (before present, with the present at 1950 C.E.). Sample spacing of the $CH_4$ measurements is between 20 - 190 years, increasing with depth. $CH_4$ concentrations vary from 355 ppb to 751 ppb. Pooled standard deviation for the measurements from 130 m to 1150m is 2.9 ppb, which considers samples both from OSU and PSU after correcting for inter-laboratory offsets. The pooled standard deviation of replicates between 1150 m and 1751 m is 2.7 ppb. The record resolves $CH_4$ signals observed in previous ice cores (Figs. 1 and 7). The mean difference between the reference $CH_4$ record from WAIS Divide (WD) and the SPC14 $CH_4$ records, determined by interpolating WD $CH_4$ data (a combination of discrete and continuous $CH_4$ measurements) to the ages of SPC14 $CH_4$ samples is 2.9 ± 1.0 ppb (one standard deviation; n = 1067), demonstrating the long-term stability of the measurement systems.

**2.2 Gas Chronology**

### 2.2.1 Summary of Synchronization Approach

To create a gas chronology for SPC14, $CH_4$ variations were visually matched at equivalent rapid $CH_4$ variations in the WD ice core; subsequently the match was optimized using an automated algorithm. The rapid changes during the last glacial period are coincident with the Northern Hemisphere Dansgaard-Oeschger (D-O) events (Baumgartner et al., 2014; Huber et al., 2006; Rosen et al., 2014; Severinghaus and Brook, 1999; Severinghaus et al., 1998) and are excellent chronostratigraphic tie points between the ice cores. The SPC14 $CH_4$ record also resolves the abrupt $CH_4$ features associated with Heinrich events, as described by Rhodes et al. (2015), and further resolves centennial scale variations in $CH_4$ previously described in the WD (Mitchell et al., 2013) and Roosevelt Island (RICE) ice cores (Lee et al., 2020), and in several records by Rhodes et al. (2017). The centennial variations are smaller in magnitude than the D-O events but are clearly present and used as Holocene tie points in the SPC19 gas chronology (Table 1, Fig. 7).

Synchronization of rapid $CH_4$ excursions between ice core records requires that both records are adequately sampled. WD was chosen as the basis for the SPC14 gas time scale because of (1) its accurate and precise chronology (WD2014) based on annual layer counting and $CH_4$ ties to Greenland ice cores and speleothem chronologies (Buizert et al., 2015; Sigl et al., 2016); (2) its high resolution continuous (Rhodes et al., 2015) and discrete (Mitchell et al. 2013; WAIS Divide Members, 2015) $CH_4$ record, minimally smoothed by gas transport in the firn (Buizert et al., 2015; Sigl et al., 2016); and (3) volcanic matching between the SP14and WD cores, providing a South Pole ice chronology synchronized to WD2014 (Winski 2019). We used the WD discrete $CH_4$ record for 0 - 9.8 ka BP, and the continuous $CH_4$ record for 9.8 ka BP until 54 ka BP (Rhodes et al., 2015; WAIS Divide Project Members, 2015). The WD2014 chronology has also been used with success for synchronization of other Antarctic ice cores (Buizert et al., 2018; Lee et al., 2020). The SPC14 $CH_4$ record has an age resolution of 25 to 150 years, which is sufficient for resolving all of the major abrupt $CH_4$ variations of the last 54,000 years as well as smaller-scale Holocene variations.

### 2.2.2 Tie Point Selection and Gas Age Uncertainty

Matching $CH_4$ variations between the WD and SPC14 records establishes the WD2014 gas age at the depth of the SP14 $CH_4$ feature being matched. Because the SP19 ice chronology has been volcanically synchronized to WD2014, this also allows us to empirically establish Δage at the depth of the $CH_4$ feature. The full gas chronology is then constructed by interpolating Δage between these tie points using a cubic spline, and subtracting this Δage spline from the ice chronology.

Tie point selection was done in two stages, first by visual matching, followed by fine-tuning of the visual match using an automated optimization algorithm. We first visually selected either the midpoint, maximum, or minimum of abrupt changes in $CH_4$, depending on the shape of the event, as tie-points. The midpoints of D-O and Heinrich $CH_4$ events were determined by averaging $CH_4$ before and after each abrupt change, then determining the midpoint between these averages, using the same techniques for averaging and defining the midpoint as described in Buizert et al., (2015). Smaller $CH_4$ variations, particularly throughout the Holocene were visually identified based on their magnitude and shape. We then optimized the tie points using a best-fit algorithm that randomly perturbs the age of each visually selected point within a 200-year window centered around the

visual tie point. The tie points were perturbed individually (i.e., one at a time). Each tie point age was randomly perturbed 1,000 times, and for each perturbation the goodness-of-fit was calculated by finding the minimum misfit, using equation (2):

$$(2)$$

$$S_m = \frac{1}{2} \sum (g_m - g_o)^2$$

where $S_m$ is the misfit, $g_m$ are the SPC14 $CH_4$ data (in ppb) after perturbing a SPC14 chronostratigraphic tie point, and $g_o$ are the WD $CH_4$ data (in ppb) we match to (i.e. the methane record of WD on the WD2014 chronology). We apply a linear interpolation to find the $g_o$ at the exact same ages as the $g_m$. We sum the squared difference at each depth (1067 total points) to calculate the misfit on each iteration of the algorithm. Once the best tie point for that event is found, the iteration is performed on the next older tie point. The automated optimization was done on high-pass filtered versions of the $CH_4$ records (1st order Butterworth with 500-
year cut off), thereby eliminating any bias created by low frequency measurement offsets. Because the methane record of the last glacial is dominated by the low to high change during the deglaciation, removing this low frequency oscillation forces the optimization algorithm to ignore this trend and only match higher frequency oscillations.

The procedure resulted in a final tie point selection where the adjustment ranged from 0.3 years to 64 years (with a mean change of 14.8 years) from the visually selected tie point, giving confidence that the matching is robust. Correlation of the WD
and SPC14 high-pass filtered records increased from r = 0.9599 (visual matching) to r = 0.9634 (automated matching). The final tie points are listed in Table 1. In the supplemental material gas ages are listed for all depths provided in the SPC14 ice age time scale data file (Winski et al., 2019) to provide unified SP19 ice age and gas age time scales for future use. Both time scales are plotted in Fig. 2. Due to the small change in tie points, the increase in correlation between the records, and the smoothly-varying reconstructed Δage, we are confident that the matching is accurate.

Three factors impact the uncertainty of the resulting gas chronology. The first is correlation uncertainty, i.e., how accurately the age of the tie point is transferred from WD to SPC14. This uncertainty is primarily controlled by the sample spacing around each tie point. The second factor is uncertainty that arises from the cubic-spline interpolation between tie points, which is more difficult to quantify. The cubic spline interpolation used here eliminates discontinuities at tie points but is not representative of the physical processes of firn densification and layer thinning. To estimate interpolation uncertainty, we examined the agreement
of small-scale methane variations between the tie points that were not explicitly matched in the procedure. Based on this evaluation the interpolation uncertainty is up to 106 years in the Holocene and up to 190 years in the glacial period. A continuous estimate of the interpolation uncertainty requires that we account for the increase in the uncertainty with distance from a tie point. Based on Fudge et al. (2014) we allow the interpolation uncertainty to increase by 10% of the age difference to the closest tie point. The third factor to consider is the absolute uncertainty of the reference WAIS Divide (WD) gas chronology (Buizert et al., 2014), which
incorporates uncertainties in the WD ice age time scale and WD Δage model. The WD chronology uncertainty changes through time based on how the chronology was created (Buizert et al., 2014). To find the estimated $2\sigma$ uncertainty along the length of the core, we used the root sum square of all three uncertainties. The uncertainties are provided in the supplement and shown in Fig. 1a.

**3 Results and Discussion**

### 3.1 An empirical record of Δage for SPC14

Accurately constraining the Δage is critical for interpreting ice core records. Traditionally, for low-accumulation Antarctic ice cores, Δage is calculated using firn densification models, as opposed to using direct gas-age, ice-age constraints (Arnaud et al., 2000; Barnola et al., 1991; Goujon et al., 2003; Loulergue et al., 2007; Lundin et al., 2017; Schwander et al., 1997), though some direct constraints on Δage do exist for Greenland ice cores (Severinghaus et al., 1998). These models simulate the physical process of firn densification over time to determine the depth and age (relative to the surface) of trapped air. Input parameters for the models (temperature, accumulation rate, surface snow density, close-off density, and in some cases dust and wind souring) as well as the physical processes involved in densification, are not known well enough in many cases, leading to substantial uncertainties in Δage when estimated through a model (Bréant et al., 2017; Freitag et al., 2013; Keenan et al., 2020). This is particularly a problem in locations or past time periods where Δage is relatively large. The difficulty in simulating past firn densification has led to uncertainties of the relative phasing of greenhouse forcing and Antarctic climate (Brook and Buizert, 2018).

SPC14 has independent ice and gas chronologies, allowing us to compute an empirically derived Δage history for SP14 with a very low relative uncertainty due to the fact that WD has a small Δage (and therefore also a small absolute Δage uncertainty). The SPC14 ice chronology was created by combining annual-layer counting with stratigraphic matching of volcanic events, and is annually resolved through the Holocene (Winski et al., 2019). The uncertainty of the Δage record is impacted by three factors, including: (1) the WD Δage uncertainty, (2) correlation uncertainty between chosen $CH_4$ tie points in the record, and (3) uncertainty in the ice age interpolation between volcanic tie points. These terms were added in quadrature to estimate a $2\sigma$ uncertainty for the empirical SPC14 Δage record, which we find increases with age (Fig. 3).

The Δage record (Fig. 3) is the first of its kind for Antarctica. It shows the expected larger Δage during the glacial period than the Holocene (due to both lower temperatures and lower accumulation rates) and an overall increase from 55 to 25 ka associated with the cooling from Marine Isotope Stage (MIS) 3 to MIS2. To assess the origin of the Holocene Δage variations, we compare our empirical Δage to firn densification model simulations results presented earlier in Winski et al. (2019). We confine this comparison to the Holocene because this section has a detailed ice chronology based on annual layer counting. Briefly, we perform three experiments using a dynamical description of the Herron-Langway densification model (Herron and Langway, 1980). In a first simulation, we force the model with realistic past accumulation variations reconstructed using the annual-layer count, and realistic past temperature variations based on the $\delta^{18}O$ of ice (isotopic slope of 0.8 ‰$K^{-1}$). A second simulation uses a constant accumulation rate (0.078 ma$^{-1}$ ice equivalent) and realistic temperature variations. A third experiment uses a realistic accumulation rate and a constant temperature (-51.5$^o$C). We find that both simulations using realistic past accumulation rates skillfully reproduce the observed variability in both $\delta^{15}N$ and Δage. By contrast, when using constant accumulation rates, the model fails to simulate the observed variations in either parameter. This is clear evidence that Holocene variations observed in our empirical Δage reconstruction are driven primarily by changes in past site accumulation rate, not site temperature. The data-model comparison of Fig.4 suggests that the Holocene section of the SP ice core, owing to its high-resolution $\delta^{15}N$ data, empirical Δage record, and annual layer count, is an ideal target for benchmarking the performance of firn densification models. The comparison shown here suggests that the dynamical version of the Herron-Langway firn model has skill in simulating past variations in firn

properties on multi-centennial time scales; whether this is true for other densification models remains to be explored (Lundin et al., 2017).

The ability of the firn model simulations to fit the $\delta^{15}N$ and $\Delta$age variations decreases towards the early Holocene. We attribute this to the fact that the model forcing is less well known as we go further back in time. Reconstructing past accumulation

requires estimates of the thinning function, which become increasingly uncertain with depth – in particular in a flank-flow configuration like SP where the deposition site moves over bedrock topography. Likewise, the temperature reconstruction becomes less certain back in time owing to corrections related to upstream elevation and isotope effects.

## 3.2 Smoothing of the SPC14 atmospheric gas record

Due to the slow firn densification process, gas diffusion and gradual bubble formation in the firn column act as a low-pass smoothing filter on the atmospheric signal (Buizert et al., 2013; Fourteau et al., 2017; Gregory et al., 2014; Schwander et al., 1993). As the firn densifies, pores remain largely open to the atmosphere, allowing the atmospheric gases to diffuse freely. At the lock in depth (LID), the firn begins to close off and diffusion of air stops (Battle et al., 1996a, 2011; Kawamura et al., 2006;

Mitchell et al., 2015). Once pore close-off occurs, no more mixing with the air above can occur. Although the impact of smoothing in the firn on gas records has long been recognized, it is not well quantified because it depends on physical processes near the firn-ice transition that are difficult and time consuming to study (Fourteau et al., 2019).

The degree to which the atmospheric signal as recorded in the ice has been filtered is of interest for understanding the speed of past environmental changes, the fidelity of the ice core gas record, and also impacts gas-to-gas correlation like the

technique employed here. For example, in a situation where an abrupt $CH_4$ increase was heavily smoothed, the damping of the concentration change (Spahni et al., 2003) would impact a tie point location. At the South Pole this issue could be a concern because this site has an unusually deep lock in depth (LID), currently ~110 m (Battle et al., 1996; Severinghaus and Battle, 2006).

To quantify the preservation of the SPC14 $CH_4$ signal at specific abrupt events we compared prominent $CH_4$ features between the SP14 and WD cores. A comparison of event duration in the WD core and the percent change in amplitude between

the event in WD and SPC14 is presented in Table 2 and Fig. 5. Event duration was determined by the number of years between the onset of rapid increases in $CH_4$ and when $CH_4$ returned to pre-event levels. As expected, our results indicate that the amplitude of shorter-lived events is reduced more than that of longer-lived events (amplitude reduction varies from 0 to 31 ppb), consistent with the findings of Spahni et al. (2003) who examined the smoothing of the 8.2 ka methane event in the EPICA Dome C ice core. However, even at values of $\Delta$age approaching 2400 years (Fig. 3), which are reached during the last glacial period, previously

identified fast $CH_4$ variations are still preserved faithfully in SP14 (Fig. 5). This level of preservation gives us confidence not only in the accuracy of the tie points, but also in how well other atmospheric gas records will be preserved in SPC14.

We apply a simple model approach to further examine how much smoothing has affected the SPC14 record. We start with the WD methane record as input, apply various smoothing filters (gas age distributions) based on a firn model, and compare the results to the SPC14 record. In doing so we assume that the WD record is a reasonable substitute for the true atmospheric history;

this assumption is justified by the high accumulation rate in WD and narrow age distribution (Battle et al., 2011; Mitchell et al., 2015; WAIS Divide Project Members, 2015).

Site smoothing is fully described by the gas age distribution in the closed bubbles. The gas age distribution employed here was created using a firn air transport model tuned to modern-day Dome C firn air sampling data and site conditions that incorporates advection, diffusion, near-surface convective mixing, deep firn dispersion and gradual bubble trapping (Buizert et al., 2012; Buizert and Severinghaus, 2016; Mitchell et al., 2015).

5    The firn air transport model was calibrated to the EDC site because it is the closest modern-day analogue to South Pole glacial conditions, with accumulation rates of around 3 cm a$^{-1}$ and a $\Delta$age of around 2,300 years. Calibration of the firn air transport model used FIRETRACC (Firn Record of Trace Gases Relevant to Atmospheric Chemical Change over 100 yrs, http://badc.nerc.ac.uk/data/firetracc) firn air sampling data of 7 atmospheric trace gases of well-known atmospheric history ($CO_2$, $CH_4$, $SF_6$, CFC-11, CFC-12, CFC-113, $CH_3CCl_3$), using established methods (Buizert et al. 2012). Bubble trapping is simulated

10  using the Mitchell et al. (2015) parameterization. Following Eq. (1) in Kohler et al. (2011), we fit a log-normal distribution to the simulated EDC age distribution; the fit is optimized using $\mu = 4.9$ and $\sigma = 0.6$. In the remainder of the analysis we shall use this log-normal distribution, as it follows precedent in the field and can be more easily replicated.

   The spectral width $\Delta$ of the gas age distribution is defined as (Trudinger et al., 2002):

$$\Delta^2 = \frac{1}{2} \int_0^\infty (t - \Gamma)^2 G(t) dt$$

with G the gas age distribution in yr$^{-1}$ and $\Gamma$ the mean of the distribution. The spectral width of the simulated EDC present-day closed-bubble gas age distribution equals 78 years (corresponding to around 3.5% of present-day EDC $\Delta$age).

   In our analysis we assume that the spectral width of the gas age distribution scales linearly with $\Delta$age, or $\Delta = \alpha \times \Delta$age;

20  where $\alpha$ is unitless scaling factor. This is a reasonable assumption one can make about the system, given that $\Delta$age represents the timescale of the snow-to-ice transformation; the gradual bubble trapping that dominates the broadening of the age distribution likely scales with this process to a large degree.

   We seek to quantify smoothing in the SPC14 $CH_4$ record by estimating the optimal scaling parameter $\alpha$. We filter the WD $CH_4$ record (assumed to reflect the true atmospheric variations) by a gas age distribution that is a linearly scaled version of

25  the simulated EDC distribution, scaled such that its spectral width reflects $\alpha \times \Delta$age at that given time in the core. We repeat this exercise for a wide range of $\alpha$ values from $1 \times 10^{-2}$ to 1 in 100 equally spaced steps. The newly filtered WD $CH_4$ record is then compared to the SP $CH_4$ record to determine the optimal value of $\alpha$ that best represents the observed degree of smoothing by minimizing a misfit function. This is illustrated in Fig. 6 (where $\alpha$ is expressed as a percentage rather than a fraction).

   The best fit to the SPC14 record uses a smoothing function history with a spectral width of 3% of $\Delta$age (or $\alpha = 0.03$).

30  This finding is of note because this amount of smoothing is much less than could be expected. In ice core science, an informal rule of thumb has been that smoothing will be 10% of $\Delta$age (Mitchell et al., 2015). This informal rule is based on the observation that the depth range of the bubble close off at many locations is about 10% of the total firn thickness. The mechanism for this small amount of smoothing requires further investigation. However, the data and our analysis show clearly that despite the large values

of Δage, significant short-term variability, including 20-30 ppb centennial scale features, will be preserved at ice core sites like the South Pole.

### 3.3 Centennial CH$_4$ variations

The SPC14 record validates previous observations of persistent centennial-scale CH$_4$ variability through the Holocene and glacial period (Mitchell et al., 2013; Rhodes et al., 2017; Lee et al., 2018) including variations matched with the WD CH$_4$ record back to 16,150 ka, just after the onset of the glacial termination (Fig. 7). Prior to 16,150 ka, similar features are not resolved in SP14 due to inadequate sampling resolution, though they have been documented in other Antarctic ice cores (Rhodes et al.,

2017). The centennial scale features observed during the Holocene are important for understanding pre-anthropogenic CH$_4$ variations. Atmospheric CH$_4$ variations in the last 2,000 years have sometimes been attributed to anthropogenic forcing mechanisms (Ferretti, 2005; Mischler et al., 2009; Sapart et al., 2012). However, recent work on the Roosevelt Island ice core (RICE) and WD ice cores and now the new SPC14 record (Fig. 7) validate the existence of similar CH$_4$ variations beginning as early as the last glacial period, well before the influence of anthropogenic forcing (Lee et al., 2018; Rhodes et al., 2017). The

observation of centennial scale variations throughout the Holocene implies that these small but consistent CH$_4$ variations occur naturally, rather than caused exclusively by human activity (Lee et al., 2020), though their origin remains unclear. Rhodes et al., (2017) hypothesized that they represent small changes in the low-latitude hydrological cycle, which lead to small changes in methane production. Whether such variations are forced or arise as a consequence of internal variability is an open question. Further work on this topic and its implication for natural CH$_4$ cycling should include further documentation of this type of

variability, investigation of source and sink variability using CH$_4$ emission and atmospheric models, and further comparison to well-dated proxies for hydroclimate at low and high latitudes. Given the large role of tropical wetlands in the modern CH$_4$ budget, exploration of centennial-scale variations in tropical hydrology may be particularly useful.

### 4.0 Summary and conclusions


The SP19 gas chronology for the SPC14 ice core covers the last 52,586 years, complementing the ice chronology presented in Winski et al. (2019). The gas chronology was created using over 2,000 high resolution, discrete CH$_4$ measurements completed at Oregon State University and Pennsylvania State University. The resulting CH$_4$ record was tied to the high resolution CH$_4$ record of the WAIS Divide ice core using the WD14 chronology. Abrupt changes in CH$_4$ at D-O events as well as distinct

variations of 20-30 ppb during the Holocene are used as tie points. The absolute uncertainty of the gas chronology changes through time to a maximum of ± 540 years at 35 ka, and an uncertainty of ± 502 years at the bottom of the core. The uncertainty is calculated to $2\sigma$ for the entire record. Key outcomes of this study include a gas age time scale for the SPC14 ice core, the observation of minimal smoothing of the gas record despite the exceptionally deep firn column at the South Pole, an empirical Δage record that can be used to test firn densification models, and the confirmation of centennial variability in atmospheric CH$_4$.


### Data Availability

The data are available in the supplementary material and the post-review time scale and data will be made fully available at the NOAA National Center for Environmental Information Paleoclimate Data base (https://www.ncdc.noaa.gov/paleo/study/31532) and the USAP Antarctic Glaciological Data Center (https://www.usap-dc.org/view/dataset/601380; https://www.usap-dc.org/view/dataset/601381).

## Author Contributions

All authors contributed data to this study. JE, EB, CB, JSE, TS, JS, EH ad MK measured ice core gases. EK and ES made isotope measurements. DW, EO, TF, KK, DF, and JK measured ice core chemistry and contributed to the ice chronology which was used

to calculate delta age. JE, EB, and CB created gas chronology. DW, TJF, DF, EK, MA oversaw the ice core collection. JE, EB, and CB wrote the manuscript with input from all authors.

## Competing Interests

The authors declare that they have no conflict of interest.

## Acknowledgements

This work was funded through grants from the US National Science Foundation (Todd Sowers (1443464 and 1804145), Edward Brook and Christo Buizert (1443472, 1643722), Erich Osterberg (1443336), Eric Steig (1143105 and 1141839), Jeffrey

Severinghaus (1443710), Murat Aydin (1142517 and 1443470), and Karl Kreutz (1443397)). We would like to thank Mark Twickler and Joe Souney with the SPC14 Science Coordination Office for their work administering the project; the U.S Ice Drilling Program for collecting the SPC14; the 109th New York Air National Guard for the airlift to and from Antarctica; the field team who collected the ice core; the members of South Pole and McMurdo stations who facilitated field operations; the National Ice Core Facility for ice core processing and storage; Ross Beaudette for his work lab work on gas datasets; and the many student

researchers who produced data for the SP19 chronologies and helped to process the core.


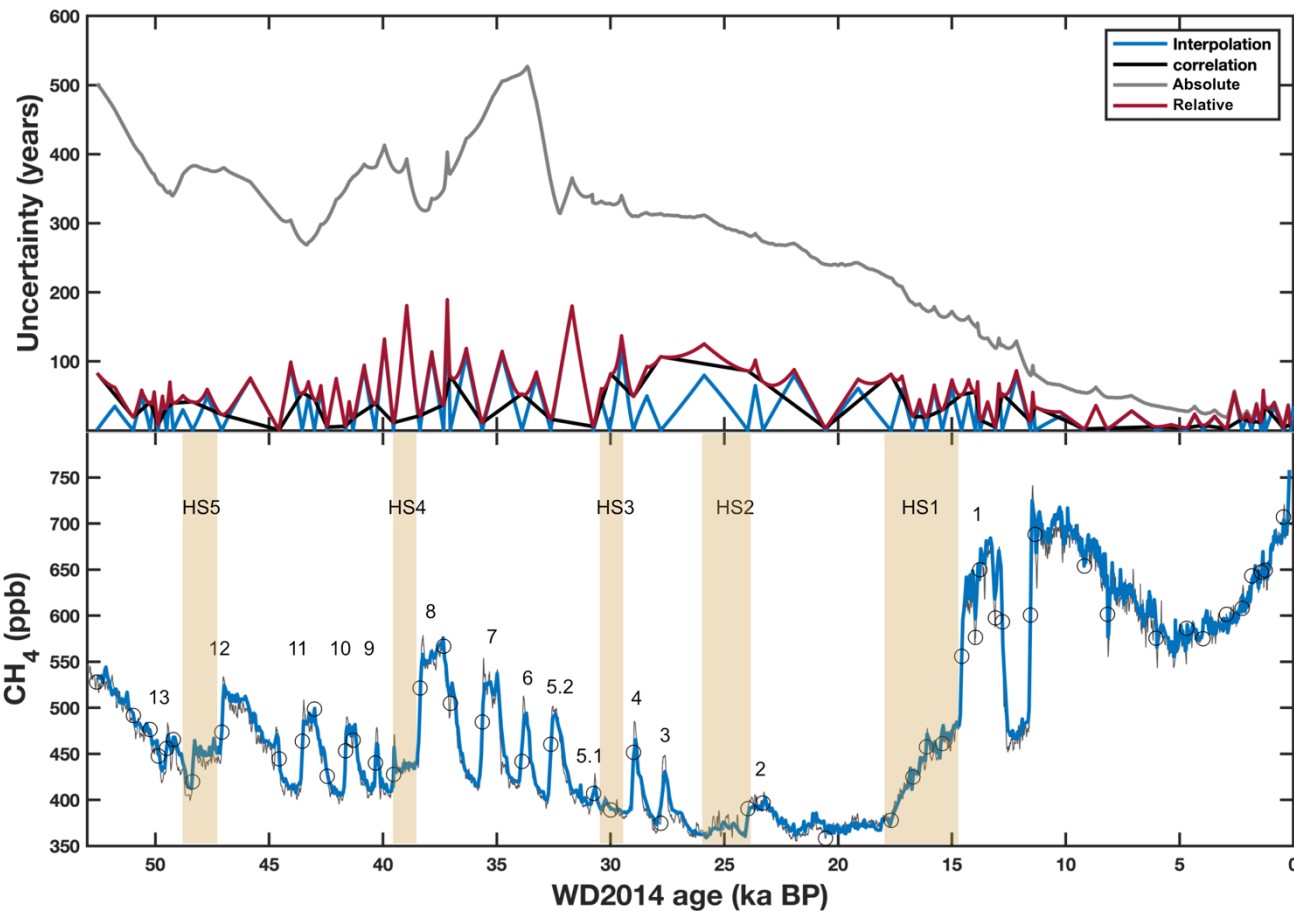

**Figure 1.** Top: SP19 gas chronology uncertainty (± 2 $\sigma$)). Black line indicates correlation uncertainty, blue line shows interpolation uncertainty, red solid line shows total SPC14 uncertainty relative to WD. The red line is a combination in quadrature of the correlation and interpolation uncertainty. The grey line describes total absolute uncertainty, which incorporates the absolute uncertainty in the WD time scale. Maximum uncertainty of ± 540 years is found around 35 ka. Bottom) SPC19 methane record (blue line), plotted on top of WD $CH_4$ record (grey) (Rhodes et al., 2015; WAIS Divide Project Members, 2015). Selected tie points are indicated by circles. The gas chronology extends from 116 years to 52,482 years BP. D-O events are numbered for orientation; yellow bars indicate Heinrich Stadials.

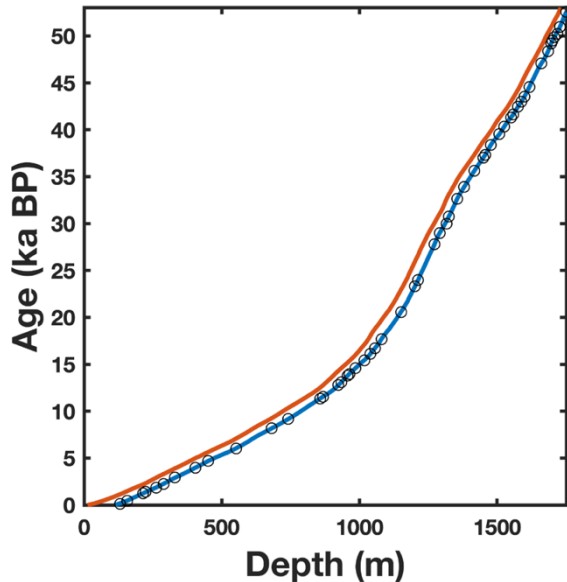

**Figure 2.** Ice age (orange) and gas age (blue) as a function of depth for the SP19 chronology. Gas tie points are indicated by black circles.

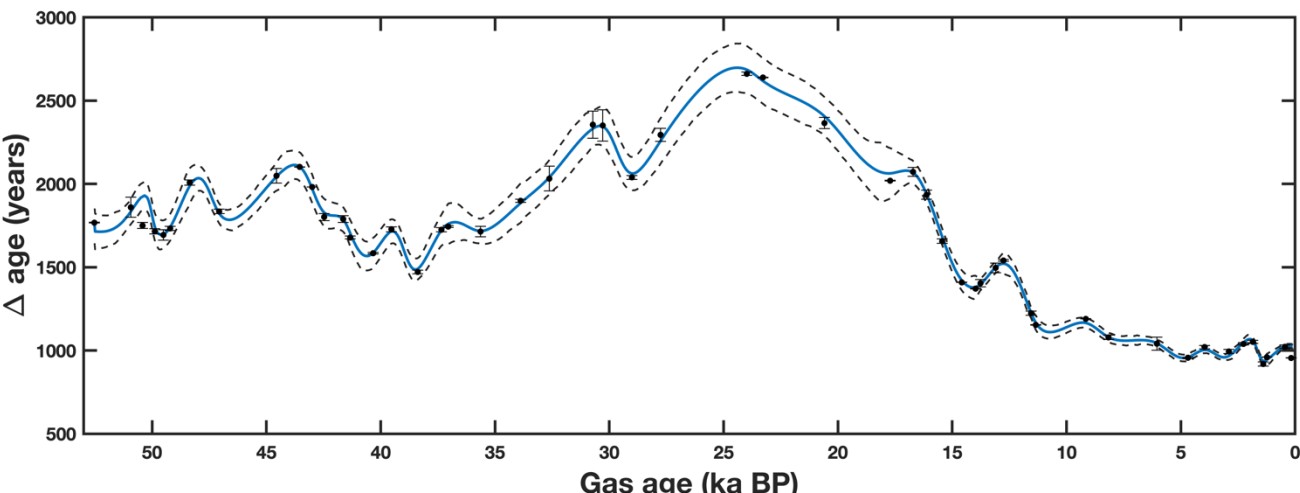

20   **Figure 3.** Empirically derived Δage history for the SPC14 ice core, derived as described in the text. Blue line is a spline fit to the Δage points. Δage error bounds (2 σ), dashed lines, reflect uncertainties with Δage based on WD Δage uncertainty, and relative SP19 uncertainties, black dots indicate individual Δage constraints (see text).

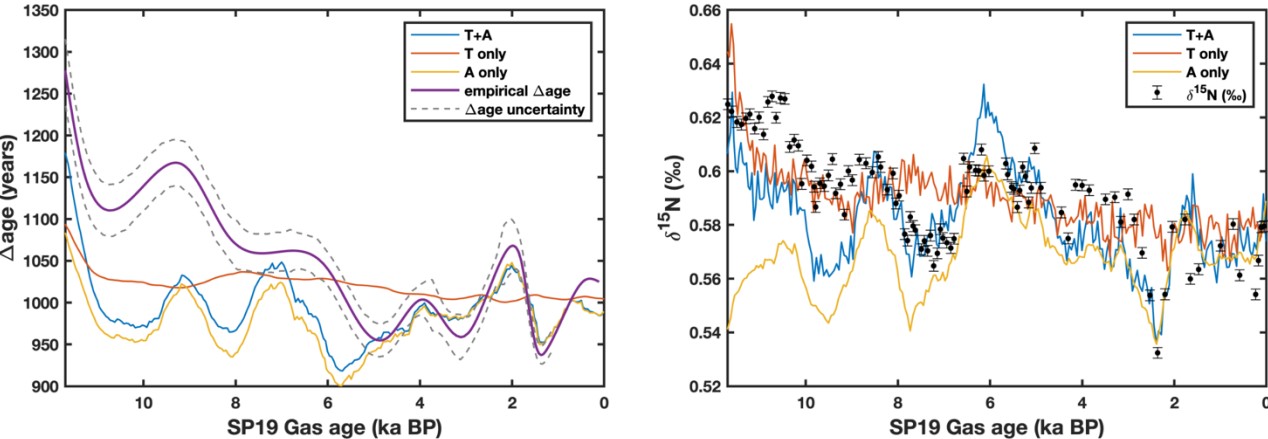

**Figure 4.** (Left) Comparison of modelled Δage (red, yellow and blue lines), and empirical Δage (purple line). Grey dashed lines represent Δage uncertainty. (Right) Modelled and actual $\delta^{15}N$ data (Winski et. al, 2019).

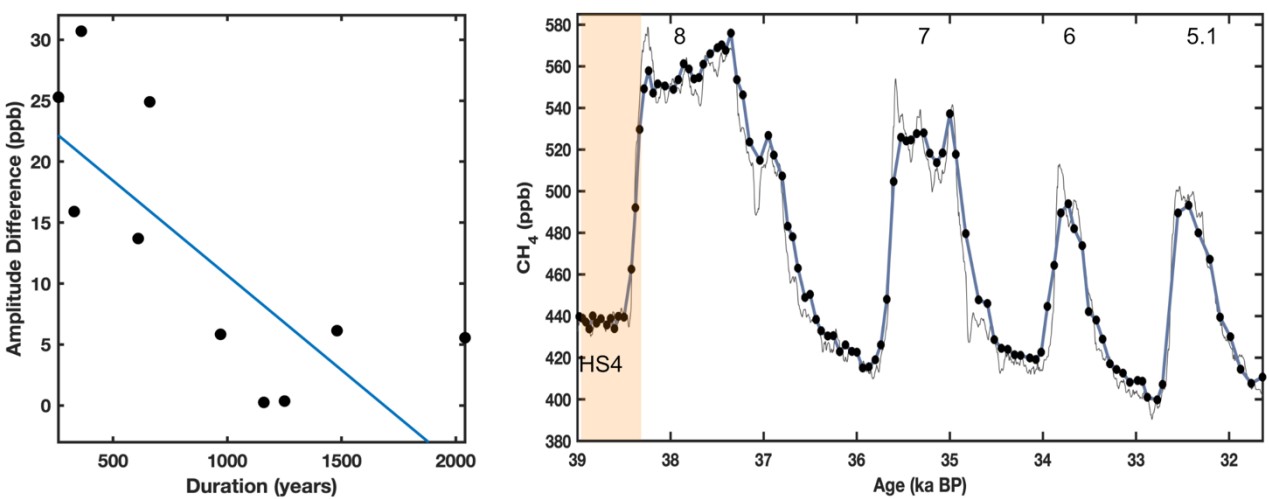

**Figure 5.** (left) Correlation of duration of event and amplitude difference between WD and SPC14 events shows a clear negative trend (r = -0.74, p = 0.006). Black markers correspond to events listed in Table 2. (right) Example of smoothing from MIS3 showing smoothing of small-scale features in SPC14 (blue) relative to WD (grey). D-O events are numbered; Heinrich Stadial 4 is shaded in orange.

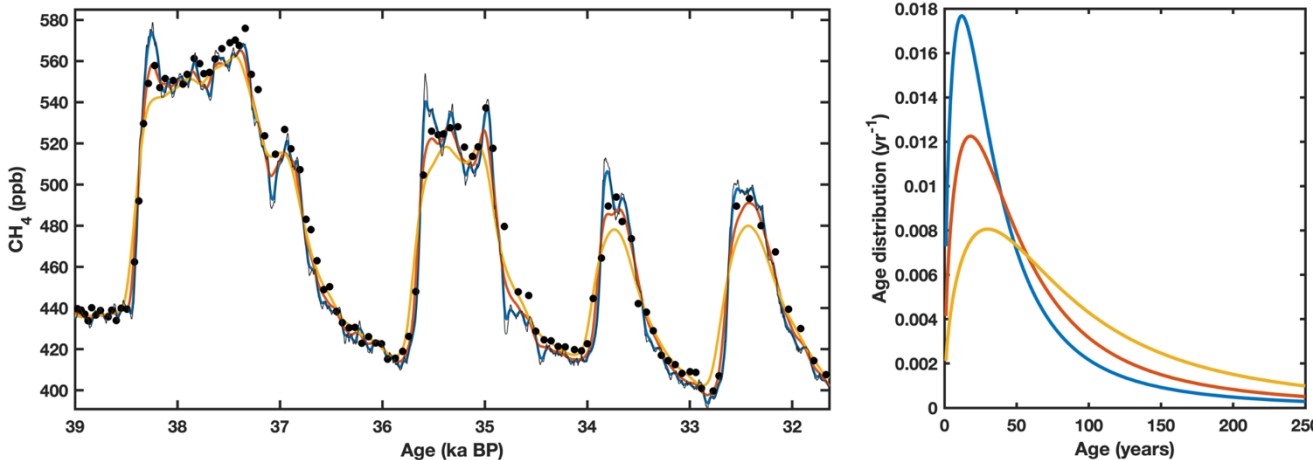

**Figure 6.** Left: SPC14 data (black dots) compared with smoothed WD CH₄ record. Colored lines show the result of smoothing the WD record with progressively wider age distributions. Original signal (grey) is plotted against three example smoothed histories: α = 0.01 (blue), α = 0.03 (red), α = 0.05 (yellow) of Δage. The best fit between the smoothed WD record and SPC14 data occurs for α=0.03 (or 3.0 % of Δage). Right: Width of the smoothing filter is defined by the spectral width, which is proportional to a percentage of Δage. Colors of the age distributions correspond with the smoothed signals in (a).

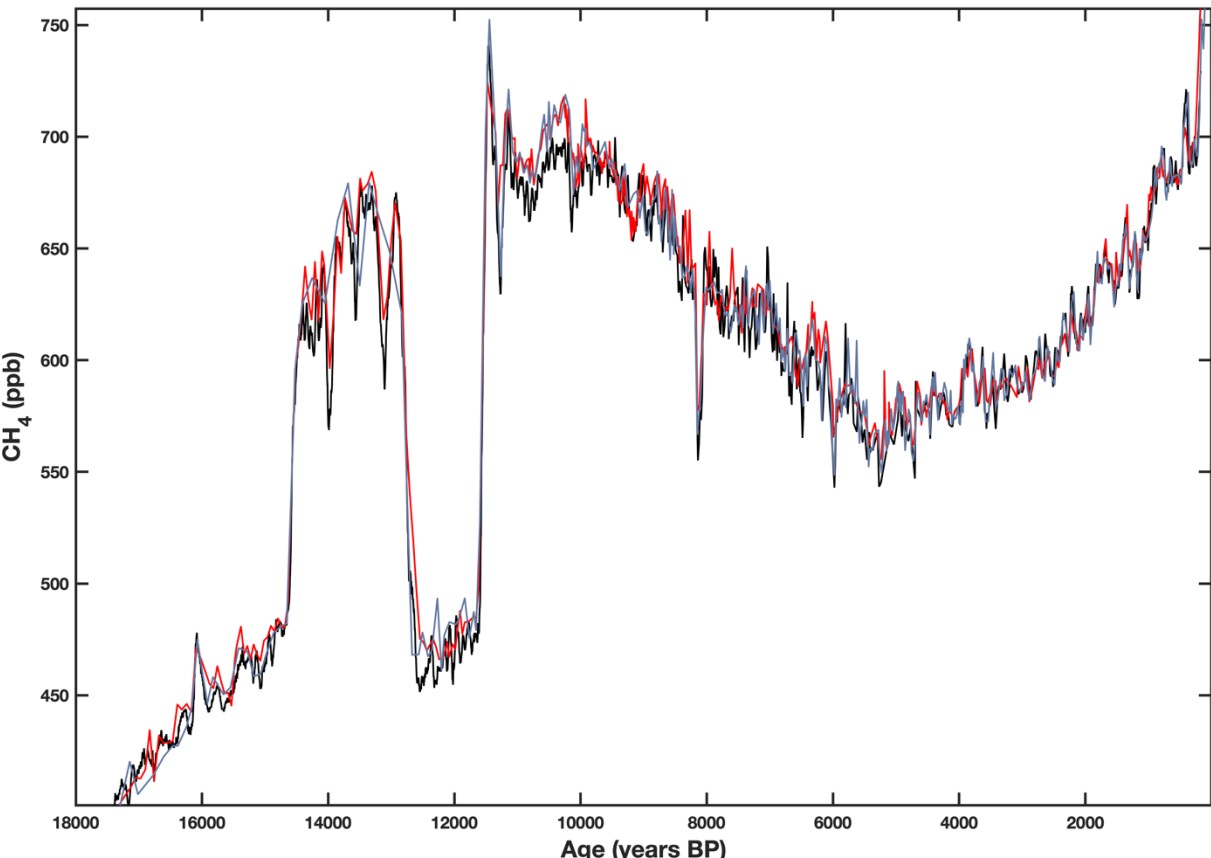

**Figure 7.** a) SPC14 (blue), WD (black), and RICE (red) all exhibit resolved centennial scale variation in $CH_4$ in the Holocene and during the deglaciation. RICE data from (Lee et al., 2020), WD data from (Rhodes et al., 2015).

**Table 1:** Tie points used for chronology with Δage and uncertainty. Uncertainties are listed at the tie points as correlation uncertainty and interpolation uncertainty. Interpolation uncertainty is given as the largest estimate for an interval between tie points. Δage is reported in years at each tie point. See supplementary material for complete time scale uncertainties. 1950 CE = 0 years.

| SPC14 Depth (m) | Gas age (yr) | Δage (yr) | Correlation uncertainty (± yr) | SPC14 Depth (m) | Gas age (yr) | Δage (yr) | Correlation uncertainty (± yr) |
|---|---|---|---|---|---|---|---|
| 130.20 | 113 | 1022 | 5.71 | 1379.45 | 33889 | 1885 | 35.95 |
| 156.25 | 443 | 1030 | 12.78 | 1417.25 | 35635 | 1716 | 36.64 |
| 214.20 | 1233 | 962 | 17.39 | 1450.00 | 37036 | 1758 | 44.60 |
| 223.60 | 1406 | 908 | 18.76 | 1457.02 | 37334 | 1731 | 31.36 |
| 261.22 | 1831 | 1071 | 15.95 | 1476.78 | 38368 | 1485 | 22.24 |
| 288.50 | 2250 | 1048 | 15.19 | 1507.95 | 39537 | 1722 | 20.82 |
| 330.00 | 2938 | 960 | 13.90 | 1525.23 | 40337 | 1580 | 23.75 |
| 403.91 | 3978 | 1007 | 13.58 | 1550.00 | 41314 | 1695 | 17.50 |
| 450.56 | 4690 | 956 | 5.75 | 1558.81 | 41640 | 1793 | 23.56 |
| 552.00 | 6039 | 1044 | 41.62 | 1575.65 | 42454 | 1821 | 23.69 |
| 681.06 | 8168 | 1079 | 14.54 | 1588.15 | 43007 | 1975 | 23.12 |
| 741.33 | 9183 | 1167 | 1.71 | 1599.17 | 43541 | 2105 | 28.32 |
| 857.20 | 11341 | 1163 | 18.69 | 1616.50 | 44564 | 2043 | 43.77 |
| 867.77 | 11547 | 1226 | 19.25 | 1660.05 | 47077 | 1822 | 25.02 |
| 923.32 | 12783 | 1522 | 31.63 | 1684.42 | 48367 | 1993 | 35.88 |
| 934.36 | 13092 | 1504 | 31.13 | 1695.79 | 49206 | 1732 | 32.47 |
| 956.36 | 13778 | 1391 | 29.25 | 1700.66 | 49503 | 1703 | 47.37 |
| 963.13 | 13969 | 1383 | 29.16 | 1707.16 | 49873 | 1723 | 19.92 |
| 985.21 | 14570 | 1419 | 26.97 | 1718.27 | 50231 | 1940 | 38.16 |
| 1017.50 | 15421 | 1676 | 27.68 | 1727.50 | 50969 | 1821 | 70.03 |
| 1039.86 | 16121 | 1937 | 37.69 | 1751.00 | 52586 | 1716 | 44.70 |
| 1056.00 | 16713 | 2070 | 36.60 | | | | |
| 1080.51 | 17677 | 2063 | 42.65 | | | | |
| 1151.00 | 20558 | 2402 | 35.12 | | | | |
| 1200.79 | 23300 | 2620 | 37.14 | | | | |
| 1212.87 | 23963 | 2685 | 53.03 | | | | |
| 1271.87 | 27798 | 2251 | 92.17 | | | | |
| 1291.23 | 28992 | 2059 | 36.09 | | | | |
| 1316.30 | 30291 | 2349 | 134.89 | | | | |
| 1324.07 | 30756 | 2328 | 83.98 | | | | |
| 1354.49 | 32624 | 2039 | 82.31 | | | | |

**Table 2: Quantification of smoothing of the methane record in SPC14.** Table shows comparison of event duration to the amplitude difference of events in the SPC14 and WD core (see Fig. 5). Spacing (in years) of SPC14 samples over the duration of the event .

| Event Name | Amplitude difference (ppb) | Percent Change (%) | Event Duration (years) | SPC14 resolution (years) |
|---|---|---|---|---|
| 1500 CE | 17.3 | 2.40 | 98 | 20 |
| YD-onset | 0.1 | 0.02 | 1250 | 37 |
| DO-3 | 16 | 3.57 | 330 | 55 |
| DO-4 | 31 | 6.40 | 360 | 40 |
| DO-5 | 25 | 5.00 | 660 | 66 |
| DO-6 | 14 | 2.73 | 610 | 47 |
| DO-7 | 6 | 1.12 | 1480 | 78 |
| DO-8 | 6 | 1.06 | 2040 | 93 |
| DO-9 | 25 | 5.26 | 260 | 52 |
| DO-10 | 5 | 1.05 | 970 | 139 |
| DO-11 | 0.2 | 0.04 | 1160 | 58 |
| DO-12 | -3 | -0.60 | 1360 | 68 |

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
