# Peer review of "The SP19 chronology for the South Pole Ice Core - Part 2: gas chronology, $\Delta$ age, and smoothing of atmospheric records"

_Climate of the Past, 2020_

## Referee Comment (RC1) · Anonymous Referee #1 · 26 Jun 2020

Epifanio et al. present the gas age chronology for the new South Pole ice core that dates back to 54 ka. The timescale is constructed by synchronization of the methane record with that of the WAIS Divide ice core following established methods. Sources of uncertainty are accounted for thoroughly. The paper is well-written and clear, and accompanied by a high-quality dataset. It will be important publication for the South Pole group and others to refer to.

I have no major issues with this straightforward study and so have listed minor comments and suggestions below. My only disappointment is that the paper touches on two interesting areas that aren't followed up: using the independent chronologies to

assess firn densification model performance and the confirmation of centennial-scale methane signals throughout the record. I look forward to subsequent manuscripts on these topics.

Abstract: Can the 'relatively minor' smoothing be quantified here? Pg 1, Line 32: Are ice core timescales really "very accurate"? Absolute age constraints are rare. Pg 2, L8: "trace impurities" – can you be more specific?

Pg 4, L1: How is the PSU blank correction "estimated"? What is the standard deviation on the 35 ppb blank? Information not in supplement. Pg 4, L15: The pooled standard deviation calculation considers samples from both labs? After the correction for the inter-lab offset? Pg 4, L18: Would be useful to know if this comparison uses WD continuous or discrete data, or combination of the two?

Pg 5, L17: What criteria were used to identify the non-DO/Heinrich tie points? Particularly through the Holocene... Pg 5,L28: Not sure I understand what "low frequency measurement offsets could cause problems for the synchronization? How was the choice of filter made? Is the filtered record displayed anywhere in figures? Doesn't using such a filter risk the introduction of signal artifacts that may bias the optimization algorithm? Pg 5, L30: 189 years is a big change – is the algorithm definitely picking out the same event in both records? Pg 5, L32: Are r-values the best method of assessing synchronization? When there are a lot of wiggles involved, as in through the Holocene, it is easy to get a high r-value while lining up the wrong events, i.e., one cycle out.

Pg 6, L10: The authors know this, but to be very clear, the WD gas age uncertainty is dependent on different things at different points in the record (i.e., Holocene vs. Glacial), depending on WD2014 construction. Pg 6, L 26: Suggest re-phrasing to "difficulty in simulating past firn densification has led to uncertainties in the relative phasing ..."

Pg 7, first paragraph. Could you provide some more information on the firn densification model results? Is the red line of Figure 4 delta-age output of one of the 3 model

results shown on Figure 13 of Winski? Why do you suggest delta-age is driven by accumulation rate changes? Pg 7, L23: "were" should be "was" Pg 8, L2: Wouldn't the WD2014 gas age dating paper be a more suitable reference here? Or just admit that WD is not atmospheric history but it's the best we've got.

Pg 8, L5: Is the "gradual bubble trapping" in model as described by Mitchell et al. 2015? Please cite if so. Pg 8, L6: Please explain how model was calibrated to EDC? Could adjustments to this tuning impact your results? Pg 8, L17: Couldn't this (gradual bubble trapping causing broadening) be demonstrated by turning off the gradual bubble trapping and comparing gas age distributions?

Pg 8, L25: Is there more discussion to have here? 3% of delta-age is less smoothing than we might have expected, at least based on the old rule of thumb of 10% delta-age. Why is the smoothing at these low accumulation rate sites less than might be expected? Pg 8, L28: Be more quantitative than "significant short term variability" Pg 8, L33: Are the centennial-scale features not present in South Pole > 16.1 ka? Or has smoothing or sampling meant they are not resolved?

Figure 1, legend. Are the labels mixed up? Figure 5, left panel. Which "events" do the black markers refer to? Surely the amplitude difference should be a percentage change to make different events comparable?

Figure 5, right panel. Please include markers for the SP data points, to show that amplitude difference is not simply a result of under-sampling. Figure 6: Maybe I missed it in the main text, why does the width of the age distribution "correspond to the median age of distribution"?

Figure 7: Could the three records be offset slightly in the y-direction to help reader see the common variations? Maybe also add a couple of sub-panels to focus in on regions of excellent match in the Holocene.

---

## Referee Comment (RC2) · Anonymous Referee #2 · 28 Jul 2020

**General comments:**

This short preprint presents a gas chronology for the South Pole 2014 ice core covering 54,000 years. It is based on a high resolution discrete methane record synchronized with the WAIS Divide methane record. In my understanding, both the methane record and the new chronology are new and important carefully built datasets that deserve to be published fast. The minor comments below intend to improve the manuscript mostly by providing more details on some aspects.

**Specific comments:**

p2 l8-10 - the study by Lee et al. (2020) about the impact of high/low impurity levels on methane records could be cited here.

p2 l14-16 and p6 l20-21 - this presentation is somewhat too simplistic, for example LIDIE estimates in AICC2012 use constraints from $\delta^{15}$N data (Veres et al., 2013; Bazin et al., 2013), and direct constraints on $\Delta$age exist at least in Greenland (e.g. Severinghaus et al., Nature, 1998)

p2 l26-28 - it would be useful to summarize how the WD gas chronology was estimated

p3 l6-8 - the intercalibration results and replicate measurements could be provided in the Supplement, illustrated by a figure and commented with more details

p3 l8 (poor sample quality) sample quality issues could be commented, with for example the brittle zone extent and ice quality.

p3 l36 - p4 l1 - an important blank correction (35ppb) due to $CH_4$ outgassing from stainless steel flasks is applied to PSU data, potentially affecting the measurement precision. I believe that the OSU and PSU data series should be provided individually in the Supplement and the intercomparison better described (see also comment on p3 l6-8).

p5 l19-27 - could the model parameters (500-year window, one by one tie point testing) influence the results? The glacial period results seem robust on Figure 1, the possibility of matching the wrong event seems less easy to exclude for the Holocene small events. This could be commented.

p6 l23-24 - some densification models use other input parameters such as dust (Freitag et al., 2013, Bréant et al., 2017) or wind (Keenan et al., 2020), this could be mentioned

p7 l5-8 - the $\Delta$age results are interesting and should be commented more extensively in order to better complement the $\delta^{15}$N based discussion in Winski et al. (2019). I think that the constraints provided by $\delta^{15}$N and the results before 8kBP in Fig. 13 of Winski et al. (2019) should be further discussed here.

p7 l26-28 and Table 2 - the $CH_4$ peak near 1500 AD, well documented in Rhodes et al. (2016) in several ice cores, also constitutes good target for smoothing evaluation, it

seems to be recorded near 153m in the South Pole data, it could be included in Table 2 and Section 3.2

p8 l3-27 - the method used to evaluate the gas age distribution characteristics leaves the reader confused about the relevance of basing the evaluation on a Dome C distribution and how it is modified by the alpha parameter. A simpler approach based on tuning a lognormal distribution could be used (e.g. Köhler et al., 2011, Fourteau et al., 2020) and would have the advantage to allow for providing the age distribution parameters (2 values) and compare them with previous estimates at other sites. Moreover, Fourteau et al. (2020) provide a high resolution EDC record of the DO6 to DO9 events, and it appears that DO9 and the $CH_4$ peak between DO9 and DO8 would provide stronger constraints to evaluate the smoothing rate of the South Pole signal and its comparison with EDC.

p9 l4-6 - this brief mention of the centennial $CH_4$ variations throughout the Holocene is interesting and could be further illustrated and commented

Figure 4 - the results before 8kaBP where the predicted $\delta^{15}N$ differs from the data in Figure 13 of Winski et al. (2019) should also be shown and commented.

**Technical Corrections:**
p2 l2 - Souney et al. (2020) is not in the list of references

p6 l29-31 - the reason why WD has small $\Delta$age could be provided for non ice core specialist readers

p7 l1 - I did not understand what is meant by "is the first of its kind" as it derived from a stratigraphic matching to WD

p7 l4 - the densification model used should be introduced

p7 l17 - the diffusion of trace gases in air does not always stop at the LID (Buizert et al., 2012) and some low accumulation sites do not show $\delta^{15}N$ plateau (Witrant et al., 2012)

p7 l27 - SP14and -> SP14 and

p9 l21-25 - I suggest to also provide a more detailed dataset including the laboratory

where the measurement originates from, corrections applied and replicate measure-ments

Figure 7 - the SPC14 record is difficult to distinguish, using a brighter colour would likely help

Table 2 - providing the depth or gas age interval would help; some Holocene events could be analysed (for example the $CH_4$ peak near 1500 AD or the 8.2 kaBP minimum studied by Spahni et al., 2003)

References not cited in the manuscript:

Bréant et al., Modelling firn thickness evolution during the last deglaciation: con-straints on sensitivity to temperature and impurities, Clim. Past, 13, 833–853, https://doi.org/10.5194/cp-13-833-2017, 2017

Fourteau et al., Estimation of gas record alteration in very low-accumulation ice cores, Clim. Past, 16, 503–522, https://doi.org/10.5194/cp-16-503-2020, 2020

Freitag et al., Impurity-controlled densification: a new model for stratified polar firn, Journal of Glaciology, Vol. 59, No. 218, doi: 10.3189/2013JoG13J042, 2013.

Keenan et al., Physics-based modeling of Antarctic snow and firn density, The Cryosphere Discussions, https://doi.org/10.5194/tc-2020-175, 2020.

Köhler et al., Abrupt rise in atmospheric $CO_2$ at the onset of the Bolling/Allerod: in-situ ice core data versus true atmospheric signals, Clim. Past, 7, 473–486, https://doi.org/10.5194/cp-7-473-2011 2011

Lee et al., Excess methane in Greenland ice cores associated with high dust concentrations, Geochimica et Cosmochimica Acta 270, 409-430, https://doi.org/10.1016/j.gca.2019.11.020, 2020.

Rhodes et al., Local artifacts in ice core methane records caused by layered bubble trapping and in situ production: a multi-site investigation, Clim. Past, 12, 1061–1077, https://doi.org/10.5194/cp-12-1061-2016, 2016

Witrant et al., A new multi-gas constrained model of trace gas non-homogeneous transport in firn: evaluation and behaviour at eleven polar sites, Atmos. Chem. Phys.,

12, 11465–11483, https://doi.org/10.5194/acp-12-11465-2012, 2012

---

## Author Comment (AC1) · 26 Aug 2020

RESPONSE TO REVIEWER 1:

*All page and line numbers refer to the originally submitted manuscript, not the corrected one.

Abstract: Can the 'relatively minor' smoothing be quantified here?

Changed [Page 1 Line 26] to "Gas transport in the firn results in smoothing of the atmospheric gas record with a smoothing function spectral width equal to 3% of delta age (ranges from 30 to 78 years)."

Pg 1, Line 32: Are ice core timescales really "very accurate"? Absolute age constraints are rare.

Changed the wording of the second sentence of the first paragraph in section 1 to read: "They are valuable recorders of past climate partly because ice cores provide detailed and well-dated the gas and ice phase, allowing comparisons to events in other ice cores and paleoarchives (Buizert et al., 2015; Elderfield et al., 2012; Hodell et al., 2017; Marcott et al., 2013)."

Pg 2, L8: "trace impurities" – can you be more specific?

Added "...such as black carbon (BC), major ions, dust, and trace elements" to line 8, giving specific examples of what 'trace impurities' are.

Pg 4, L1: How is the PSU blank correction "estimated"?

Added to page 4 line 5: "The blank was estimated by analyzing ice of a known CH4 concentration through multiple melt-refreeze cycles. A regression between the excess CH4 and the number of melt-refreeze cycles was completed to arrive at an estimate of 35 ±19 ppb blank correction. This correction was applied to all PSU samples."

What is the standard deviation on the 35 ppb blank? Information not in supplement.

Standard Deviation of the PSU blank correction is ± 19 ppb. Information was added to the supplement, "CH4 Blank Corrections" tab and to Page 4 line 3.

Pg 4, L15: The pooled standard deviation calculation considers samples from both labs? After the correction for the inter-lab offset?

The pooled standard deviation does consider samples from both labs; however, they are differentiated by depth because PSU measured ice above 1150m. From 130-1150m, the pooled standard deviation considers all the measurements from both labs, after correction for the interlab offset. For example, samples measured at the same depth but at both labs were considered one group, and their standard deviation was

used to calculate the pooled standard deviation. Deeper than 1150 m, measurements were all conducted at OSU, and have a different pooled standard deviation. To make this clearer, "Pooled standard deviation for the measurements is 2.9 ppb between 130 m to 1150 m, and 2.7 ppb between 1150 m – 1751 m." on page 4, line 18 was changed to "Pooled standard deviation for the measurements from 130 m to 1150 m is 2.9 ppb, which considers samples both from OSU and PSU after correcting inter-laboratory offsets. The pooled standard deviation from 1150 m to 1751 m is 2.7 ppb. Pooled standard deviation breakdown by laboratory is given in the supplement."

Pg 4, L18: Would be useful to know if this comparison uses WD continuous or discrete data, or combination of the two?

This comparison uses a combination of the WD discrete and continuous data, the same combination that is referred to later on Page 5, Line 3. WAIS divide data were internally consistent between discrete and continuous measurements. Added: "(a combination of discrete and continuous CH4 measurements)" after "...determined by interpolation WD CH4 data" on Pg 4 line 18.

Pg 5, L17: What criteria were used to identify the non-DO/Heinrich tie points? Particularly through the Holocene...

Non-DO/ tie points were visually identified. Added "non-DO events, particularly through the Holocene were visually identified based on magnitude and shape of the small event." Pg 5 Line 19, after sentence that ends "...as described in Buizert et al., (2015)."

Pg 5, L28: Not sure I understand what "low frequency measurement offsets could cause problems for the synchronization?

"Because the methane record of the last glacial is dominated by the low to high change during the deglaciation, removing this low frequency oscillation forces the optimization algorithm to ignore this trend and only match higher frequency oscillations." Added as the last sentence on page 5 line 29.

How was the choice of filter made?

Filter was made to ensure it removed low-frequency signals from the CH4 record. See attached figure.

Is the filtered record displayed anywhere in figures?

Not currently, but the figure is attached to this response for the editor and/or reviewers to evaluate. We do not think it adds much to the manuscript, and have left it out for that reason – but we would be happy to include it if requested to do so.

Doesn't using such a filter risk the introduction of signal artifacts that may bias the optimization algorithm?

We think that signal artifacts are avoided based on the cutoff frequency of the filter, and based on how the filtered signal looks, as shown in the attached figure. Timing of the abrupt changes is not shifted by the filtering. We plotted the filtered vs unfiltered data, and saw no artifacts that could bias the optimization algorithm.

Pg 5, L30: 189 years is a big change – is the algorithm definitely picking out the same event in both records?

The 189-year change occurred on one event, around 30,000 years (tie point number 29). We reviewed the algorithm, and narrowed the window for tie points to move from ±250 years to ±100 years, and came up with a new tie point, that visually looked like a much better fit, and also removed an abrupt change in delta age, giving further confidence that this was a better fit. The explanation for this is that the algorithm cannot distinguish between similar looking events (duration and magnitude) that are relatively close. No other tie points changed significantly after narrowing the tie point freedom, so we remain confident that the age scale is correct.

The gas age scale and empirical delta age was updated appropriately in the supplement, and Table 1. The new tie point 29 is now located at 30,291 ka BP.

Pg 5, L32: Are R-values the best method of assessing synchronization When there are a lot of wiggles involved, as in through the Holocene, it is easy to get a high R-value while lining up the wrong events, i.e., one cycle out?

R values do an adequate job in showing a change (improvement) in the record after optimization. Other metrics (such as the root mean square deviation) would give very similar results. We considered the chance of getting similar results based on lining up the wrong events, but the events are different enough that there would not be an improvement. Additionally, close visual inspection convinces us that the record is correctly lined up. Finally, most of the small events in the Holocene are moved less than 20 years, where cycles between events in the Holocene are centennial scale or larger. This allows us to be confident that the events are matched correctly. Another line of evidence that the match is correct are the smooth changes in empirical delta age. Added "Due to the small change in tie points, the increase in correlation, and the smoothly varying empirical Delta-age , we are confident that the matching is accurate." To the end of pg 5, line 35.

Pg 6, L10: The authors know this, but to be very clear, the WD gas age uncertainty is dependent on different things at different points in the record (i.e., Holocene vs. Glacial), depending on WD2014 construction.

Added to page 6, line 11.: "The origin of the WD chronology uncertainty changes through time based on how the chronology was created (Buizert et al., 2014)."

Pg 6, L 26: Suggest re-phrasing to "difficulty in simulating past firn densification has led to uncertainties in the relative phasing ..."

Changed, as suggested.

Pg 7, first paragraph. Could you provide some more information on the firn densification model results?

Added to page 7, line 3: "To assess the origin of the Holocene $\Delta$age variations, we

compare our empirical $\Delta$age to firn densification model simulations results presented earlier in Winski et al. (2019). Briefly, we perform three experiments using a dynamical description of the Herron-Langway densification model (Herron and Langway 1980). In a first simulation, we force the model with realistic past accumulation variations reconstructed using the annual-layer count, and realistic past temperature variations based (isotopic slope of 0.8 ‰-1); in a second simulation a constant accumulation rate (0.078 m a-1 ice equivalent) and realistic temperature variations; in a third experiment a realistic accumulation rate and a constant temperature (-51.5oC). We find that both simulations using realistic past accumulation rates skillfully reproduce the observed variability in both d15N and ïĄĎage during the last ∼5ka. By contrast, when using constant accumulation rates, the model fails to simulate the observed variations in either parameter. This is clear evidence that Holocene variations observed in our empirical $\Delta$age reconstruction are driven primarily by changes in past site accumulation rate, not site temperature. The data-model comparison of Fig.4 suggests that the late-Holocene section of the SP ice core, owing to its high-resolution d15N data, empirical D-age record, and annual layer count, is an ideal target for benchmarking the performance of firn -densification models. The comparison shown here suggests that the dynamical version of the Herron-Langway firn model has skill in simulating past variations in firn properties on multi-centennial time scales; whether this is true for other densification models remains to be explored (Lundin et al. 2017, FIRN model inter-comparison)."

Is the red line of Figure 4 delta-age output of one of the 3 results shown on Figure 13 of Winski? Changed Figure 4 to include all the delta age outputs (3 model runs) to match Winski et al (2019), and to include the d15N model outputs, changed caption to read:

"Figure 4. (Left) Comparison of modelled $\Delta$age (red, yellow and blue lines); see text for details) and empirical $\Delta$age (purple line). Grey dashed lines represent $\Delta$age uncertainty. (Right) Modelled and actual d15N data (Winski et. al, 2019)."

Why do you suggest delta-age is driven by accumulation rate changes?

We have now clarified this point by presenting three separate firn model simulations that were described in response to an earlier comment. We show that using realistic accumulation variations (as derived from the annual layer count) to force the firn model is both a necessary and sufficient requirement to simulate the observed variations in both ïĄĎDage and d15N.

Pg 7, L23: "were" should be "was":

changed as suggested.

Pg 8, L2: Wouldn't theWD2014 gas age dating paper be a more suitable reference here? Or just admit that WD is not atmospheric history but it's the best we've got.

Mitchell et al, 2015 was referenced here because of the work that described the small bubble close off and narrow age distribution in the WD ice core. We added the WD2014 gas age dating paper also, as suggested by the reviewer.

Pg 8, L5: Is the "gradual bubble trapping" in model as described by Mitchell et al. 2015? Please cite if so.

Yes. "Mitchell et al., 2015" added to list of references on P8 L15.

Pg 8, L6: Please explain how model was calibrated to EDC? Could adjustments to this tuning impact your results?

In response to reviewer comments, we have re-evaluated the smoothing using a log-normal age distribution as suggested by Kohler et al. 2011. This change did not impact our result.

Removed from Page 8, Line 6: "The model was calibrated to the EDC site because it is the closest modern-day analogue to South Pole glacial conditions, with accumulation rates of around 3 cm a-1 and a △age of around 2300 years. The resulting age distribution is similar to that obtained by Spahni et al. (2003) using a similar approach."

And Replaced on Page 8 line 6: "The firn air transport model was calibrated to the

EDC site because it is the closest modern-day analogue to South Pole glacial conditions, with accumulation rates of around 3 cm a-1 and a △age of around 2300 years. Calibration of the firn air transport model was done using FIRETRACC (Firn Record of Trace Gases Relevant to Atmospheric Chemical Change over 100 yrs, http://badc.nerc.ac.uk/data/firetracc) firn air sampling data of 7 atmospheric trace gases of well-known atmospheric history ($CO_2$, $CH_4$, $SF_6$, CFC-11, CFC-12, CFC-113, $CH_3CCl_3$), using established methods (Buizert et al. 2012). Bubble trapping is simulated using the Mitchell et al. (2015) parameterization. Following Eq. (1) in Kohler et al. (2011), we fit a log-normal distribution to the simulated EDC age distribution; the fit is optimized using ïĄ■ = 4.9 and ïĄş=0.6. Using the lognormal distribution makes it easier for future studies to apply our findings to other sites."

Pg 8, L17: Couldn't this (gradual bubble trapping causing broadening) be demonstrated by turning off the gradual bubble trapping and comparing gas age distributions?

All existing closed porosity parameterizations have gradual bubble trapping, and so this feature is not specific to Mitchell et al. 2015; that study merely studies the effect of density layering (rather than gradual bubble trapping) on the age distribution. It is not possible to turn off gradual bubble trapping in the firn air transport model.

Pg 8, L25: Is there more discussion to have here? 3% of delta-age is less smoothing than we might have expected, at least based on the old rule of thumb of 10% deltaage. Why is the smoothing at these low accumulation rate sites less than might be expected?

Changed the last paragraph of section 3.2 to read: "The best fit to the SPC14 record uses a smoothing function history with a spectral width of 3% of △age (or a = 0.03). This amount of smoothing is much less than what could be expected from an informal rule of thumb that smoothing should be about 10% of △age (Mitchell et al., 2015). This informal rule is based on the observation that the depth range of the bubble trapping at many locations is about 10% of the total firn thickness. The mechanism for this

small amount of smoothing requires further investigation, but is probably linked to the cessation of vertical diffusion at the lock-in depth which effectively eliminates vertical gas motion relative to the ice matrix prior to full bubble occlusion. Our results show that despite the large values of delta age, significant short-term variability, such as 20-30 ppb centennial scale features, can be preserved at ice core sites like the South Pole."

Pg 8, L28: Be more quantitative than "significant short-term variability"

added, 'including small 20-30 ppb centennial scale features' after 'significant short term variability.

Pg8, L33: Are the centennial-scale features not present in South Pole > 16.1 ka? Or has smoothing or sampling meant they are not resolved?

Centennial scale features before this time period are not resolved by sampling. Added "Prior to 16,150 ka, similar features are unable to be resolved due to sampling resolution" on pg 8 line 33.

Figure 1, legend. Are the labels mixed up?

The labels are mixed up, thanks for catching this. Figure 1 legend changed to grey line = Absolute, Red line = Relative.

Figure 5, left panel. Which "events" do the black markers refer to?

Black markers refer to events listed in Table 2. Added "black markers refer to events in Table 2" to caption of Figure 5

Surely the amplitude difference should be a percentage change to make different events comparable?

Added a percentage change column to the Table 2. Also, changed "A comparison of event duration in the WD core and the difference in amplitude between the event in WD and SPC14 is presented in Table 2 and Fig. 5." On page 7, line 27 to "A comparison of event duration in the WD core and the percent change in amplitude between the event

in WD and SPC14 is presented in Table 2 and Fig. 5." This change does make the comparison more accurate, and does not change our results.

Figure 5, right panel. Please include markers for the SP data points, to show that amplitude difference is not simply a result of under-sampling. Data points added:

Figure 6: Maybe I missed it in the main text, why does the width of the age distribution "correspond to the median age of distribution"?

Deleted this line from caption 6, line 4 as it adds confusion. The point was only to draw the reader's attention to the shape of the age distribution becoming wider, and causing more smoothing. Without this line, the reader can see that the shape of the filters based a percentage of delta age cause progressively more smoothing in the record.

RESPONSE TO REVIEWER 2

p2 l8-10 - the study by Lee et al. (2020) about the impact of high/low impurity levels on methane records could be cited here.

Added citation "Lee et al, 2020" after Aydin et al., 2004 on pg 2 line 10.

p2 l14-16 and p6 l20-21 - this presentation is somewhat too simplistic, for example LI-DIE estimates in AICC2012 use constraints from _15N data (Veres et al., 2013; Bazin et al., 2013), and direct constraints on dage exist at least in Greenland (e.g. Severinghaus et al., Nature, 1998).

Added to p2 line 16: "Later Antarctic chronologies introduced constraints on age scales using d15N, d18Oatm and firn models (Bazin et al., 2012; Veres et al., 2013)."

Changed to p6 line 21: "Traditionally, for low-accumulation, Antarctic, ice cores, $\Delta$age is calculated using firn densification models, as opposed to using direct gas-age, ice-age constraints (Arnaud et al., 2000; Barnola et al., 1991; Goujon et al., 2003; Loulergue et al., 2007; Lundin et al., 2017; Schwander et al., 1997), there are some direct constraints on delta age do exists for Greenland ice cores (Severinghause et al., 1998)."

p2 l26-28 - it would be useful to summarize how the WD gas chronology was estimated.

Added to page 2, Line 28, second sentence " The WD2014 ice core chronology was created in two parts (Buizert 2015, Sigl 2016): from 0-31.2 ka BP it is based on annual-layer counting; from 31.2-67.8 ka BP it is based on stratigraphic matching of WD CH4 to NGRIP d18O (using a linear correction of the NGRIP age scale to improve the fit to the Hulu speleothem record). For both segments the WD2014 ïĄĎage estimate is based on an d15N-constrained firn densification model simulation.

p3 l6-8 - the intercalibration results and replicate measurements could be provided in the Supplement, illustrated by a figure and commented with more details.

Lab intercalibration is given in a figure below. It is put, along with OSU and PSU data in supplement, under tab "Lab Comparison". Figure caption reads as below:

Figure S1: Lab comparison of CH4 measurements. Difference was calculated by de-trending the measurements, averaging each lab's CH4 values, and determining the difference. PSU measurements were determined to be 6 ppb lower than OSU mea-surements, and were corrected by adding 6 ppb to PSU measurements for the duration of the record.

p3 l8 (poor sample quality) sample quality issues could be commented, with for exam-ple the brittle zone extent and ice quality.

Poor sample quality issues were exclusively due to being broken in shipment, and the brittle ice zone did not affect our study. Changed "poor sample quality" to" samples that had been broken during shipment." On pg3 line 8.

p3 l36 - p4 l1 - an important blank correction (35ppb) due to CH4 outgassing from stainless steel flasks is applied to PSU data, potentially affecting the measurement precision. I believe that the OSU and PSU data series should be provided individually in the Supplement and the intercomparison better described (see also comment on p3 l6-8).

Inter-comparison is given in a figure attached to supplement, figure S1, with a caption that describes the OSU/PSU inter-comparison. PSU and OSU raw data are given in supplement.

p5 l19-27 - could the model parameters (500-year window, one by one tie point testing) influence the results? The glacial period results seem robust on Figure 1, the possibility of matching the wrong event seems less easy to exclude for the Holocene small events. This could be commented.

As we mentioned in our comments to reviewer 1, R-values do an adequate job in show-ing a change (improvement) in the record after optimization. We considered the chance of getting similar results based on lining up the wrong (i.e. adjacent) events, but the se-lected events are different enough that such scenarios can easily be rejected. Finally, most of the small events in the Holocene are moved less than 20 years, so moving over one cycle did not occur, even though we gave the algorithm enough freedom to move beyond events. Added "Due to the small change in tie points and the increase in correlation between the records, we are confident that the matching is accurate." To the end of pg 5, line 35.

p6 l23-24 - some densification models use other input parameters such as dust (Freitag et al., 2013, Bréant et al., 2017) or wind (Keenan et al., 2020), this could be mentioned.

Added "…and in some cases dust loading and/or wind scouring)" to p6 l24 to end of list of model inputs. Also added suggested citations at end of sentence.

p7 l5-8 - the _age results are interesting and should be commented more extensively in order to better complement the _15N based discussion in Winski et al. (2019). I think that the constraints provided by _15N and the results before 8kBP in Fig. 13 of Winski et al. (2019) should be further discussed here.

We now plot both the d15N and Dage for each of the three simulations of Winski et al. (2019), and compare these to the observations; this was described earlier in the

response to reviewer 1. We also show the results further back in time, and added more discussion.

p7 l26-28 and Table 2 - the CH4 peak near 1500 AD, well documented in Rhodes et al. (2016) in several ice cores, also constitutes good target for smoothing evaluation, it seems to be recorded near 153m in the South Pole data, it could be included in Table 2 and Section 3.2.

Added this peak to the analysis, added to Table 2. This changed the statistics to r = -0.74, p = 0.006. This was updated in the caption of figure 5.

p8 l3-27 - the method used to evaluate the gas age distribution characteristics leaves the reader confused about the relevance of basing the evaluation on a Dome C distribution and how it is modified by the alpha parameter. A simpler approach based on tuning a lognormal distribution could be used (e.g. Köhler et al., 2011, Fourteau et al., 2020) and would have the advantage to allow for providing the age distribution parameters (2 values) and compare them with previous estimates at other sites. Moreover, Fourteau et al. (2020) provide a high resolution EDC record of the DO6 to DO9 events, and it appears that DO9 and the CH4 peak between DO9 and DO8 would provide stronger constraints to evaluate the smoothing rate of the South Pole signal and its comparison with EDC.

"Following the reviewer suggestion, we now use a lognormal distribution, and have edited the text as noted above, in comments to reviewer 1 (Page 8, Line 6). As noted by the reviewer, this makes it easier to compare to previous work, and makes it easier to apply the method to other sites. In response to comparing the SP data to the EDC DO events presented in Fourteau (2020), we think it is outside the scope of the paper to present the issue in detail beyond the characteristics of the South Pole record, though we find the observation interesting."

p9 l4-6 - this brief mention of the centennial CH4 variations throughout the Holocene is interesting and could be further illustrated and commented.

We think that the discussion is adequate for the scope of the paper, which is to present the gas chronology and point out observations of the record such as the degree of smoothing. We are working on a more comprehensive analysis of this variability, which does not appear (so far) to have a very simple origin in a recognized climate forcing. We would prefer not to speculate further at this point.

Figure 4 - the results before 8kaBP where the predicted _15N differs from the data in Figure 13 of Winski et al. (2019) should also be shown and commented.

The figure will be updated, to show data for the entire Holocene.

Added as the last paragraph of section 3.1: "The ability of the firn model simulations to fit the d15N and Dage variations decreases towards the early Holocene. We attribute this to the fact that the model forcing is less well known as we go further back in time. Reconstructing past accumulation requires estimates of the thinning function, which become increasingly uncertain with depth – in particular in a flank-flow configuration like SP where the deposition site moves over bedrock topography. Likewise, the temperature reconstruction becomes less certain back in time owing to corrections related to upstream elevation and isotope effects."

[Figure]

**Fig. 1.** Filtered CH4 record

[Figure]

**Fig. 2.** Updated Figure 4

[Figure]

**Fig. 3.** Updated Figure 5

[Figure]

**Fig. 4.** Figure S1